# ZeroSlide: Is Zero-Shot Classification Adequate for Lifelong Learning in Whole-Slide Image Analysis in the Era of Pathology Vision-Language Foundation Models?

Doanh C. Bui[✉], Hoai Luan Pham, Vu Trung Duong Le, Tuan Hai Vu, Van Duy Tran, Yasuhiko Nakashima

Nara Institute of Science and Technology, Japan
`bui.cao_doanh.bd2@naist.ac.jp`

**Abstract.** Lifelong learning for whole-slide images (WSIs) poses the challenge of training a unified model to perform multiple WSI-related tasks, such as cancer subtyping and tumor classification, in a distributed, continual fashion. This is a practical and applicable problem in clinics and hospitals, as WSIs are large, require storage, processing, and transfer time. Training new models whenever new tasks are defined is time-consuming. Recent work has applied regularization- and rehearsal-based methods to this setting. However, the rise of vision-language foundation models that align diagnostic text with pathology images raises the question: *are these models alone sufficient for lifelong WSI learning using zero-shot classification, or is further investigation into continual-learning strategies needed to improve performance?* The empirical study demonstrates that a well-pretrained pathology vision-language foundation model, when used with a simple zero-shot approach, can achieve competitive performance compared to training-based rehearsal and regularization-based continual learning methods. To our knowledge, this is the first study to compare conventional continual-learning approaches with vision-language zero-shot classification for WSIs. Our source code and experimental results will be available at `https://github.com/caodoanh2001/ZeroSlide`.

**Keywords:** lifelong learning · whole slide image analysis · pathology vision-language foundation model.

## 1 Introduction

Whole-slide images (WSIs) are gigapixel in size and provide visualization of tissue at the cellular level, playing a key role in cancer diagnosis and prognosis [1]. Computational tools have been developed to support diagnostic tasks such as cancer subtyping [2], tumor classification [3], [4], cancer grading [5], [6], and survival analysis [7], [8]. However, the rapid growth in WSI volume has led to an increasing number of related tasks. Moreover, because WSIs are so large, they require substantial storage, processing, and transfer time. Therefore, it is

necessary to investigate how to extend a unified computational model to new WSI-related tasks without retraining or building a new model to save time and effort.

Prior studies on lifelong learning primarily fall into two categories: regularization- and rehearsal-based methods [9]–[14]. Regularization-based methods constrain the parameters learned on the current task to remain close to those of previous tasks. Notable examples include LwF [9] and EWC [10]. Rehearsal-based methods maintain a fixed-size buffer of representative samples from past tasks for replay during new-task training; examples include ER-ACE [12], AGEM [13], and DER++ [14]. In WSI analysis, ConSlide [15] introduced BuRo, a buffer strategy that partitions slides into regions and randomly recombines them to diversify the buffer without increasing its capacity. Subsequently, [16] proposed a distance consistency loss that minimizes the discrepancy between pairwise distances of current replay sample representations and those stored in a memory bank, thereby stabilizing the replay queue.

Concurrently, MI-Zero [17] introduced a similarity computation between vision features and class prompts for zero-shot classification of WSI tasks following self-supervised contrastive learning, yielding promising results. Foundation pathology vision-language models, such as CONCH [18] and TITAN [19], have been developed using self-supervised methods to align slide embeddings with diagnostic text. These models further strengthen zero-shot classification by matching pathology visual features with text prompts.

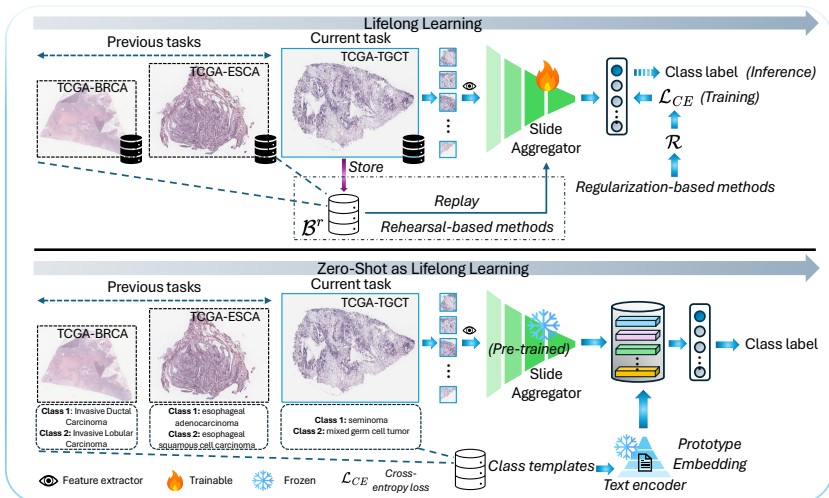

**Fig. 1.** Regularization-based and rehearsal-based methods require retraining when adding tasks, while zero-shot classification with a pathology vision-language model only needs new class templates, making it training-free. This study compares the performance of lifelong learning with training-free zero-shot classification to training-based continual learning methods.

Given these developments, we pose the following research question: *Is zero-shot classification sufficient for lifelong learning on WSIs when leveraging an advanced pathology vision-language foundation model, or are additional continual learning techniques required?* If zero-shot classification is treated as a lifelong learning method, adding a new task requires only defining a new class text prompt and using a pathology-specific vision-language foundation model to extract text embeddings as a classifier, rather than performing time-consuming continual learning. In this study, we frame zero-shot classification as a lifelong learning approach and compare it head-to-head with training-based continual learning methods on WSI analysis tasks to determine whether zero-shot classification suffices or if continual learning methods still offer superior performance. Our experiments, designed to answer this question, reveal that zero-shot classification is *highly competitive* with continual-learning-based models.

## 2   Experimental Designs

### 2.1   Problem Definition

We define the lifelong learning problem for WSI analysis as follows. Let $\mathcal{D} = \{D_i\}_{i=1}^{N}$ be a sequence of $N$ tasks or datasets. Each $D_i$ is partitioned into $D_i = D_i^{\text{train}} \cup D_i^{\text{test}}$, where $D_i^{\text{train}}$ is used for training and $D_i^{\text{test}}$ for evaluation. After training on the $t$-th task using $D_t^{\text{train}}$, the algorithm $\mathcal{F}$ must maintain its performance on $\{D_i^{\text{test}}\}_{i<t}$, minimizing forgetting as much as possible. Our objective is to evaluate: (1) whether **zero-shot classification** alone suffices to develop $\mathcal{F}$, or: (2) whether **continual learning approaches**, where $\mathcal{F}$ is trained with techniques designed to mitigate forgetting, are required. Following the evaluation settings of continual learning studies [13]–[15], there are two scenarios: **class-incremental (CLASS-IL)** and **task-incremental (TASK-IL)**. CLASS-IL requires the model to correctly predict the true class label across all accumulated classes as the number of tasks grows, whereas TASK-IL considers only the logits for the current task's classes. Hence, CLASS-IL is more challenging.

### 2.2   WSI Tiling & Feature Extraction

Given a WSI, we tile it into $K$ patches using the segmentation and patching strategy of CLAM [3]. We then use the vision-language foundation model TITAN's vision encoder [19] to extract features for each patch. This yields a sequence of patch features $\mathbf{x} = \{x_i\}_{i=1}^{K}$, where $x_i \in \mathbb{R}^{C_{\text{vis}}}$ is a $C_{\text{vis}}$-dimensional feature vector. The sequence $\mathbf{x}$ then undergoes a slide aggregation function $f_{\mathcal{A}}$ to obtain a single slide embedding for classification. To incorporate continual learning, $f_{\mathcal{A}}$ is learnable, and we use HIT [15], which leverages the pyramid structure of a WSI. For zero-shot classification, $f_{\mathcal{A}}$ is pretrained and requires no further training when adapting to new tasks.

### 2.3   Lifelong Learning for WSIs using Continual Learning-based Models

**For the regularization-based model**, we include Elastic Weight Consolidation (EwC) [10], which leverages parameters from the $(t-1)$-th task's model $\theta_{t-1}$ to regularize training of the current model $\theta_t$. EwC [10] approximates the posterior importance of each parameter by a Gaussian centered at $\theta_{t-1}$ with precision given by the diagonal of the Fisher information matrix $F$, then adds a quadratic penalty to prevent important parameters from drifting. These techniques mitigate forgetting by constraining updates to directions deemed critical for previous tasks.

**For rehearsal-based models**, we select Dark Experience Replay (DER++), a widely used rehearsal-based method, and ConSlide [15], specifically designed for continual learning on WSIs. DER++ performs knowledge distillation by aligning the prediction logits of the current model $f_t^{(i)}$ at iteration $i$ with those of a past model $f_t^{(k)}$ for $k < i$, thereby reducing forgetting. ConSlide proposes a hierarchical transformer to leverage the pyramid structure of WSIs and introduces the BuRo strategy, which breaks a WSI into regions, stores them in a buffer, and then randomly merges them to form new WSIs for replay. For continual learning-based methods, the slide aggregator $f_{\mathcal{A}}$ is trainable and includes a classification head to generate logits.

### 2.4   Lifelong Learning as WSI Zero-Shot Classification

To formulate lifelong learning as zero-shot classification, we first create a set of class templates for each test dataset $D_i^{\text{test}}$. Following the zero-shot setup of [17], we define $T = 22$ base templates (e.g., ''`a histopathological image showing [CLASS].`") and generate $\approx 4$ phrasing variants per class. For a dataset with $m$ classes, this yields about $4m$ sentences per class and a total of $\approx 88m$ prompts. We denote the $j$th class's prompts collectively as $c_j$, and set $C_i = \{c_j\}_{j=1}^m$. Each prompt in $C_i$ is fed into a text encoder of TITAN vision-language foundation model $f_{\text{text}}^{\text{TITAN}}$, and we average the resulting embeddings across variants to obtain one prototype embedding per class.

**ZeroSlide: Adapt To Lifelong Learning.** Algorithm 1 details how these embeddings are used for zero-shot classification in the lifelong learning setting. First, a global set of prototypes $\mathcal{T}$ is defined. For each new $i$-th task, we define its class templates $C_i$ and obtain their embeddings via $f_{\text{text}}^{\text{TITAN}}$. Given test patch features $\mathbf{x}_j \in D_i^{\text{test}}$, we aggregate them into a slide embedding $s_i$ and compute its similarity to each prototype. The prototype with the highest similarity to $s_i$ determines the predicted cancer subtype. In the CLASS-IL scenario, $s_i$ is compared against all prototypes in $\mathcal{T}$; in the TASK-IL scenario, it is compared only to the prototypes of the current task.

We refer to the strategy to adapt zero-shot classification as lifelong learning as **ZeroSlide**.

---

**Algorithm 1** ZeroSlide: Lifelong Learning as WSI Zero-Shot Classification

---

**Input**: Sequence of test datasets $\mathcal{D}^{\text{test}} = \{D_i^{\text{test}}\}^{|\mathcal{D}|}$, class templates $\mathcal{C} = \{C_i\}_{i=1}^{|\mathcal{D}|}$
**Output**: Predicted class label
**Initialize**: Set of prototype embeddings $\mathcal{T} = \varnothing$
**Supporting Operations**: TITAN text encoder: $f_{\text{text}}^{\text{TITAN}}$, pre-trained slide encoder: $f_{\mathcal{A}}$

1: **for** $D_i^{\text{test}}$ **in** $\mathcal{D}^{\text{test}}$ **do**
2:      $T_i \leftarrow \left\{ f_{\text{text}}^{\text{TITAN}}(c_k) \mid c_k \in C_i \right\}_{k=1}^{|C_i|}$
3:                          ▷ Extract text embeddings for the $i$-th task, where $T_i \in \mathbb{R}^{|C_i| \times \dim}$
4:      $\mathcal{T} \leftarrow \mathcal{T} \cup T_i$          ▷ Add to the text-based classifier, where $\mathcal{T} \in \mathbb{R}^{\sum_{k \leq t} |C_k| \times \dim}$
5:      **for** $\mathbf{x}_j$ **in** $D_i^{\text{test}}$ **do**
6:          $s_j \leftarrow f_{\mathcal{A}}(\mathbf{x}_j)$          ▷ Aggregate the set of patch features, where $s_j \in \mathbb{R}^{1 \times \dim}$
7:          $\hat{p}_j^{CI} \leftarrow s_j \odot \mathcal{T}^{\mathsf{T}}$ ▷ Compute CLASS-IL similarity, where $\hat{p}_j^{CI} \in \mathbb{R}^{1 \times \sum_{k \leq t} |C_k|}$
8:          $\hat{p}_j^{TI} \leftarrow s_j \odot T_i^{\mathsf{T}}$          ▷ Compute TASK-IL similarity, where $\hat{p}_j^{TI} \in \mathbb{R}^{1 \times |C_i|}$
9:          $\hat{y}_j^{CI} \leftarrow \arg\max \hat{p}_j^{CI}$                          ▷ Get CLASS-IL prediction
10:        $\hat{y}_j^{TI} \leftarrow \arg\max \hat{p}_j^{TI}$                          ▷ Get TASK-IL prediction

---

# 3  Experiments

## 3.1  Datasets

We establish a sequence of six TCGA datasets: TCGA-BRCA (breast), TCGA-RCC (kidney), TCGA-NSCLC (lung), TCGA-ESCA (esophagus), TCGA-TGCT (testis), and TCGA-CESC (cervix uteri). Each dataset addresses a cancer subtyping task. The dataset details are shown in Fig. 2. Each dataset is split into 10 folds, each comprising a train–validation–test split. All experiments are run on 10 folds to ensure stability. For continual learning–based models, training is performed on $D_i^{\text{train}}$; checkpoints are saved based on performance on the validation set $D_i^{\text{val}}$, and results are reported on $D_i^{\text{test}}$. For ZeroSlide, only $D_i^{\text{test}}$ is used for evaluation using Algorithm 1 without training.

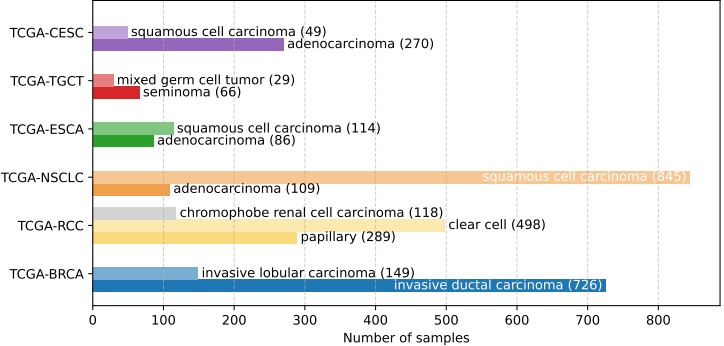

**Fig. 2.** Distribution of six TCGA datasets.

### 3.2   Metrics

There are five metrics: Accuracy (ACC), Masked Accuracy (MASKED ACC), Mean Accuracy (mACC), Backward Transfer (BWT), and Forgetting. ACC measures performance under the CLASS-IL scenario using prediction logits $\hat{p}_j^{CI}$ computed with all accumulated prototypes. MASKED ACC measures performance under the Task-IL scenario using prediction logits $\hat{p}_j^{TI}$ computed with prototypes of the current task only. The mACC is the running average of mean task accuracies: after each new task $i$, compute the mean accuracy over the $i$ tasks seen so far and then average these means across the sequence. BWT quantifies how learning new tasks affects past tasks, indicating positive or negative transfer. Forgetting measures knowledge loss by comparing the highest accuracy achieved on task $i$ with its final accuracy after training on all $|\mathcal{D}|$ datasets/tasks.

### 3.3   Implemental Details

For all models, we train 10 epochs per task sequentially on six TCGA datasets using the same random seed to ensure stable comparisons. For the backbone $f_\mathcal{A}$ used to extract slide embeddings in all continual learning methods, we employ HIT [15], which is designed to aggregate features from the patch to region level. Regions are tiled at $10\times$ magnification into $1024 \times 1024$ pixel areas and then each region is cropped into $4 \times 4$ patches of $256 \times 256$ pixels. Both patch and region features are obtained using $f_{\text{vis}}^{\text{TITAN}}$, with $C_{\text{vis}} = 768$. For embedding dimension in HIT, we use $dim = 384$.

### 3.4   Experimental Results

**Main Results.** The results, reported in Tab. 1, reveal an interesting finding: despite being training-free, *ZeroSlide achieves competitive performance with rehearsal-based continual learning methods DER++ and ConSlide, and significantly outperforms the regularization-based method EWC.* For rehearsal-based methods, with a buffer size of $|\mathcal{B}_r| = 30$, ConSlide attains the highest ACC in the CLASS-IL scenario (65.673 %) and the second-best Masked ACC in the TASK-IL scenario (90.255 %), as well as mACC (82.602 %). These margins over ZeroSlide are small (+1.544 % ACC, +0.497 % Masked ACC, and +0.009 % mACC). Furthermore, ZeroSlide outperforms DER++ (with $|\mathcal{B}_r| = 10$ or 30) in ACC by +5.131 % and +7.510 %, respectively, although DER++ still leads in TASK-IL. We also observe that *ZeroSlide is more stable with respect to the BWT and Forgetting metrics.* Regarding Forgetting, ZeroSlide achieves the best score (0.909), while securing the second-best BWT (–0.909). These results suggest that advanced training-based WSI-specific continual learning models still outperform ZeroSlide in CLASS-IL accuracy. However, adapting zero-shot classification to lifelong learning is both promising and feasible, as ZeroSlide's performance remains not only highly competitive with training-based continual learning methods under both CLASS-IL and TASK-IL scenarios but also demonstrates the most stability in BWT and Forgetting metrics.

**Table 1.** Experimental results on a sequence of six TCGA datasets. Red highlights the best performance, while blue highlights the second-best.

| Method | Buffer size $|\mathcal{B}_r|$ | ACC | MASKED ACC | mACC | BWT ↑ | Forgetting ↓ |
|---|---|---|---|---|---|---|
| _Regularization-based Methods (Training-based)_ | | | | | | |
| EWC | 0 | 43.522 ($\pm$6.765) | 89.244 ($\pm$1.902) | 69.834 ($\pm$3.566) | -3.617 ($\pm$3.101) | 4.649 ($\pm$2.558) |
| _Rehearsal-based Methods (Training-based)_ | | | | | | |
| DER++ | | 56.619 ($\pm$4.027) | 89.571 ($\pm$1.051) | 81.786 ($\pm$1.575) | -3.627 ($\pm$1.182) | 4.266 ($\pm$1.089) |
| ConSlide | $\approx 10$ | 64.226 ($\pm$4.282) | 89.318 ($\pm$2.349) | 81.992 ($\pm$1.055) | 0.196 ($\pm$1.745) | 4.849 ($\pm$2.527) |
| DER++ | | 58.998 ($\pm$1.219) | 90.604 ($\pm$1.472) | 83.580 ($\pm$1.532) | -2.368 ($\pm$1.889) | 3.199 ($\pm$1.640) |
| ConSlide | $\approx 30$ | 65.673 ($\pm$1.780) | 90.255 ($\pm$1.334) | 82.602 ($\pm$1.201) | -2.930 ($\pm$1.506) | 4.032 ($\pm$1.248) |
| _Lifelong Learning as Zero-Shot Classification (Training-free)_ | | | | | | |
| ZeroSlide | 0 | 64.129 ($\pm$1.591) | 89.758 ($\pm$0.984) | 82.593 ($\pm$0.017) | -0.909 ($\pm$0.406) | 0.909 ($\pm$0.406) |

**Confidence Score Study.** We examine the prediction results of all models to investigate the stability of predictions across tasks after training the final task, as shown in Fig. 3. We analyze the prediction scores corresponding to the ground-truth labels to assess the model's confidence in the true labels. For ZeroSlide, we consider all scores computed with prototypes in $\mathcal{T}$. For the other three models, we examine the logits across all cancer subtypes in $|\mathcal{D}|$ datasets/tasks after softmax. Our first observation is that ZeroSlide's score is significantly lower than the other continual learning models. This is expected, as ZeroSlide is training-free and uses class template prototypes while distance between the slide embedding and the prototype is not perfectly close. For the training-based models, the score with the target label is high. However, EWC shows significant degradation in confidence for TCGA-BRCA (median $\approx 0.1$) and TCGA-RCC (median $\approx 0.3$) as tasks increase, while DER++ and ConSlide maintain high performance on these datasets (median $\geq 0.75$). All models struggle with TCGA-ESCA test samples, with extremely low confidence scores. EWC and DER++ even achieve a median confidence score of 0 on TCGA-ESCA, while ZeroSlide has a median of $\approx 0.09$. EWC and DER++ also show low confidence for TGCT. Overall, ConSlide demonstrates the best stability in confidence as tasks increase. While ZeroSlide's scores with target prototypes are modest, they are still sufficiently sensitive to correctly classify cancer subtypes, making its performance highly competitive with ConSlide and other continual learning models.

## 4    Discussion

Based on the experimental designs and results, we conclude that _leveraging advanced pathology vision-language foundation models and performing zero-shot_

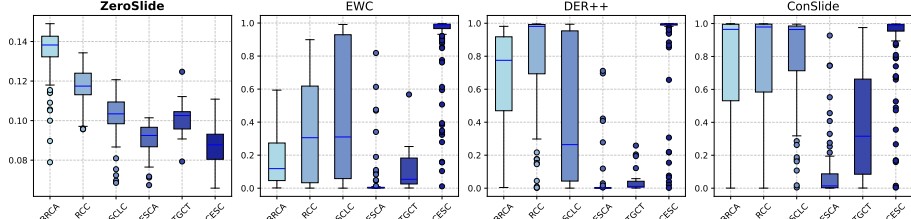

**Fig. 3.** Confidence scores for target cancer subtype labels after training/inference on the final tasks of ZeroSlide and all continual-learning-based models.

*classification is feasible for lifelong WSI analysis* in two key ways: 1) ZeroSlide's performance asymptotically matches or surpasses continual learning methods, and 2) it is fully training-free, requiring no storage for WSIs. Despite these promising results, we highlight the limitations of ZeroSlide and suggest future improvements for lifelong learning.

**Risk of Ambiguous Class Prediction.** As shown in Fig. 3, ZeroSlide's confidence score for the target label is significantly lower than that of DER++ and ConSlide. This indicates that if class templates are poorly defined or if tasks include out-of-distribution class names, ZeroSlide may not perform effectively in a lifelong learning setting.

**Perspectives for Improvement.** Although ZeroSlide is effective, training-based continual learning methods still yield strong results. Future work could involve integrating class templates with methods like ConSlide to enhance performance. The CATE approach [20] maximizes informative features using text prompts but requires time and storage for class template embeddings during inference. A promising direction would be to limit class prompt usage to online training, ignoring it during inference. Additionally, current continual learning methods require training over multiple epochs when new tasks are added. An ideal approach would involve leveraging class templates from pathology vision-language foundation models to minimize training epochs.

## 5    Conclusion

This study examines zero-shot classification using a pathology vision-language foundation model (ZeroSlide) and compares it with training-based continual learning methods for lifelong WSI analysis. Experimental results across six TCGA datasets suggest that ZeroSlide performs similarly to continual learning models, while being training-free and incurring no storage or online buffer costs. However, some limitations are noted, and future studies are recommended to improve lifelong learning for WSIs. We believe this study bridges the knowledge gap in zero-shot classification within the pathology vision-language model era and encourages developments to make lifelong learning for WSI analysis more applicable and practical in clinical settings.

## Acknowledgments

This work was supported in part by the Japan Science and Technology Agency (JST)-Advanced Technologies for CArbon-Neutral (ALCA-Next)-Next Program, Japan, under Grant JPMJAN23F4; in part by the Japan Society for the Promotion of Science (JSPS), Grants-in-Aid for Scientific Research (KAKENHI), Japan, under Grant 22H00515; and in part by the Next Generation Researchers Challenging Research Program under Grant zk25010020.

## Disclosure of Interests

The authors declare that they have no conflict of interest.

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
