# OpenReview forum: "ZeroSlide: Is Zero-Shot Classification Adequate for Lifelong Learning in Whole-Slide Image Analysis in the Era of Pathology Vision-Language Foundation Models?"
_MICCAI.org/2025/Workshop/MSB_EMERGE — MSB EMERGE 2025 Oral_

### Official Review · Reviewer_gsjt · 2025-07-07

**Recommendation:** 5
**Confidence:** 4

**Clarity:**

The paper is clear and well-written, with minor areas for improvement in clarity

**Feedback:**

- **Clarification of metrics**: The definitions and calculations of the reported metrics, particularly Backward Transfer (BWT) and Forgetting, should be explained more clearly. It is currently unclear how these values are computed and whether the reported scores are generally considered good or poor, beyond their relative comparison across methods.
- **Missing results on independent tasks**: The results suggest that some of the tasks are more challenging than others (ACC significantly lower than Masked ACC and mACC). Providing a breakdown of performance by individual task would offer a deeper understanding of task-specific difficulty.
- It appears that the BWT and Forgetting scores for ZeroSlide are nearly identical. The authors are encouraged to double-check these values to ensure they are correct.

**Justification:**

The authors present a strategy for WSI classification that generalizes across tasks without requiring additional training. The paper is well written, and the results support the effectiveness of ZeroSlide, making it an interesting contribution to the community.

**Reproducibility:**

Sufficient amount of details available for reproducing the main results, and open access is provided (or promised upon acceptance) to source code and/or data

**Strengths:**

- **High performance**: The results show the effectiveness and efficiency of vision-language approaches for zero-shot WSI classification.
- **Extensive benchmark**: The paper uses 6 datasets to evaluate the continual learning and zero-shot learning strategies.

**Summary:**

The authors demonstrate that integrating text embeddings with image features improves performance in scenarios where multiple tasks are learned sequentially. The proposed approach outperforms regularization-based continual learning methods and remains competitive with rehearsal-based strategies. In addition, it achieves these results without the need for additional training.

**Weaknesses:**

- **Missing fully-supervised baseline**: Table 1 shows that there is forgetting across tasks for all methods, and most methods show negative backward transfer. Including a fully-supervised baseline trained on all datasets simultaneously would provide essential context for evaluating the performance of continual learning methods and better highlight the advantages (or limitations) of the proposed approach compared to standard training strategies.

---

### Official Review · Reviewer_aXFA · 2025-07-08

**Recommendation:** 4
**Confidence:** 3

**Clarity:**

The paper is clear and well-written, with minor areas for improvement in clarity

**Feedback:**

* The abstract could better highlight the key result (ZeroSlide performs comparably to continual learning baselines without training).
* It might be helpful to include organ-wise analysis to provide a more detailed understanding of the method’s behavior across datasets.

**Justification:**

The paper is clearly written and presents a meaningful contribution. The results are encouraging, and the simplicity of the approach makes it appealing for real-world use. Overall, this is a valuable step toward leveraging foundation models in pathology under realistic constraints.

**Reproducibility:**

Sufficient amount of details available for reproducing the main results, and open access is provided (or promised upon acceptance) to source code and/or data

**Strengths:**

* The ZeroSlide framework is simple, intuitive, and easy to apply in practice.
* It shows that a zero-shot approach can be competitive with continual learning methods, which is a promising and encouraging result.

**Summary:**

This paper presents ZeroSlide, a simple zero-shot framework for lifelong learning in pathology. Without any additional training, it uses a frozen vision-language foundation model and achieves performance comparable to continual learning methods.

**Weaknesses:**

* While the method is tested on multiple TCGA datasets, a breakdown of performance by organ would make the analysis more informative.
* Only one foundation model (TITAN) is evaluated. It would be helpful to see results from other models like CONCH to confirm generality.

---

### Official Review · Reviewer_s7ZS · 2025-07-08

**Recommendation:** 4
**Confidence:** 3

**Clarity:**

The paper is clear and well-written, with minor areas for improvement in clarity

**Feedback:**

- I would have assumed that metrics such as forgetting should be a perfect score for zero-shot prediction as there is no training involved and the method should always perform the same. Where are those changes in the metrics coming from?
- Zero-shot prediction performance can be very sensitive to changes in the prompts. I wonder how different prompting strategies could impact the performance.

**Justification:**

The paper presents an interesting approach at continual learning. It is simple and elegant and I believe the paper is a good contribution to the community with lots of directions to explore the idea further.

**Reproducibility:**

Sufficient amount of details available for reproducing the main results, and open access is provided (or promised upon acceptance) to source code and/or data

**Strengths:**

- The idea presented in the paper is an elegant and simple way to approach continual learning without actually having to do any training, by simply generating prompts for new tasks.
- Competitive performance of the method against training approaches, the method is nicely evaluated against regularization and rehearsal based method.

**Summary:**

The paper introduces ZeroSlide, an approach using the zero-shot prediction capabilities of pre-trained pathology vision-language foundation models, such as CONCH and TITAN, as as a way of lifelong learning in whole-slide image analysis. The idea is evaluated against a number of training based continual approaches, including rehearsal and regularization based methods.

**Weaknesses:**

- There is a need for a better description of the metrics. Looking at the forgetting metric, I’d have assumed that it should achieve a perfect score for an approach involving no training where there should be no change in the results of the tasks and no forgetting. I believe this part might need a bit more clear explanation, explaining the nuances of why this could happen. It also seems to have a negative backward transfer which I’d think should be absolutely neutral?
- While the zero-shot approach should be able to generalize to new tasks, it removes the possibility that further training for new tasks can actually improve the performance of the past tasks.
- The zero-shot performance is linked to the model it is performed on. Using different vision-language models can yield very different results and this hasn’t been discussed in the paper.